# Multi Mycotoxin Determination in Dried Beef Using Liquid Chromatography Coupled with Triple Quadrupole Mass Spectrometry (LC-MS/MS)

**DOI:** 10.3390/toxins12060357

**Published:** 2020-05-29

**Authors:** Toluwase Adeseye Dada, Theodora Ijeoma Ekwomadu, Mulunda Mwanza

**Affiliations:** 1Department of Animal Health, Faculty of Natural and Agricultural Sciences, Mafikeng Campus, North West University, Private Bag X2046, Mmabatho 2735, Mafikeng, South Africa; 23115394@nwu.ac.za; 2Ekiti State College of Agriculture and Technology, Isan Ekiti 371106, Ekiti State, Nigeria

**Keywords:** aflatoxins, mycotoxins, liquid chromatography, mass spectrometry, ochratoxin A, solid-liquid extraction, zearalenone, dried beef

## Abstract

Dried beef meat, a locally processed meat from the cow, is vulnerable to contamination by mycotoxins, due to its exposure to the environmental microbiota during processing, drying, and point of sale. In this study, 108 dried beef samples were examined for the occurrence of 17 mycotoxins. Samples were extracted for mycotoxins using solid-liquid phase extraction method, while liquid chromatography coupled with triple quadrupole mass spectrometry (LC-MS/MS) via the dilute and shoot method was used to analyze the mycotoxins. Aflatoxin was found in 66% of the samples (average value of 23.56 µg/kg). AFB_1_ had a mean value of 105.4 µg/kg, AFB2 mean value of 6.92 µg/kg, and AFG_1_ and AFG_2_ had an average mean value of 40.49 µg/kg and 2.60 µg/kg, respectively. The total aflatoxins exceed the EU (4 μg/kg) permissible level in food. The α-Zea average mean value was 113.82 µg/kg for the various selling locations. Also, cyclopiazonic acid had an average mean value of 51.99 µg/kg, while some of the beef samples were contaminated with more than nine different mycotoxins. The occurrence of these mycotoxins in dried beef is an indication of possible exposure of its consumers to the dangers of mycotoxins that are usually associated with severe health problems. This result shows that there are mycotoxin residues in the beef sold in Ekiti State markets.

## 1. Introduction

Mycotoxins contamination in food has remained a major concern in food safety as they are toxic secondary metabolites produced by several molds and often contaminate food and feed worldwide [1]. Owing to the likely natural contamination of grains used by the food and livestock industry, there is a high probability of mycotoxins to enter the food chain [2,3]. Their incidence is influenced by various factors, such as the food product type, climatic conditions, agricultural practices, storage conditions, and periodic variances [4,5]. Mycotoxins have the potential to affect human health by acute and chronic effects, such as the induction of hepatocellular carcinoma (HCC) or unexpected death due to acute toxicity in the case of aflatoxins [6].

Prior research has revealed that products of animal sources, for example meat and meat products, can also increase the human mycotoxins’ intake coming either from indirect transfer from farm animals exposed to naturally contaminated grains and feed (carry-over effects) or direct contamination with fungi, or naturally contaminated seasoning blends used in meat production [7,8]. Among mycotoxins of health importance in relation to their toxicity and incidence, aflatoxin (AFs), ochratoxin A (OTA) fumonisins (FBs), and zearalenone (ZEN) are the most frequently encountered contaminants [9]. AFB_1_ is the most potent liver carcinogen recognized in mammals, classified by the International Agency for Research on Cancer (IARC) as a group 1 carcinogen, OTA in group 2B, and ZEN in group 3 carcinogens [10].

The occurrence of mycotoxins in food and feed is potentially hazardous to the health of humans in addition to animals, because of their carcinogenic, toxic, and mutagenic properties [11]. Among farm animals, pigs are known to be predominantly sensitive to mycotoxins, while ruminants such as sheep and cows are less predisposed, as their rumen fluid is capable of enzymatically degrading these mycotoxins into less toxic metabolites [7]. It has been stated [12] that the immune modulation effects of some mycotoxins increase health impacts of major illnesses disturbing Africa, such as malaria, kwashiorkor, and HIV/AIDS. It is apparent that sub-Saharan residents are prone to prolonged dietary mycotoxin exposure since they regularly eat affected crops, and due to the fact that crops in tropical and subtropical areas are more vulnerable to contamination owing to the suitable climatic conditions [13,14].

A number of studies have shown that fungi species belonging to *Penicillium* and *Aspergillus* genera are producers of these mycotoxins, and have been isolated from meat products, such as processed sausages or dry-cured hams [15,16,17]. There are other studies that showed that significant OTA levels are likely to be found in meat products produced from contaminated raw materials [3,8], as well as in smoked meat and meat products [18,19]. It is also acknowledged that aflatoxins can be found in meat as well as in meat products, if the animals eat sufficient amounts of AFB_1_ occurring in their feed [15,20,21].

In Nigeria, the consumption of dried meat is increasing especially in the south-west region and there are studies indicating the presence of fungi capable of producing mycotoxins on the dried meat from other states in the south-west region [22,23]. This might be as a result of the preservation method that is usually employed to lower the water activity such as dehydration, and the addition of salt in some cases. However, as the meat surfaces dry, there is increased possibility of growth of undesirable fungi, several of which can produce potent mycotoxins, due to the favorable conditions during the drying, storage, and selling process. The presence of fungi on regular food items in Nigeria is not in doubt, since the prevalence of mycotoxins in sub-Saharan Africa has been reported [24,25]. However, there is a dearth of information on multi-mycotoxins in dried beef in Nigeria while a lot of attention is placed on other food products. In lieu of the above facts, the present study is aimed at investigating the level of AFs, OTA, and ZEN using liquid chromatography coupled with triple quadrupole mass spectrometry (LC-MS/MS) to assess a wider range of potential mycotoxins that contaminates dried meat sold in Ekiti State destined for human consumption.

## 2. Results and Discussion

### 2.1. Method Validation

The results obtained from the validation method used in the sample extraction and LC-MS method in the determination of mycotoxins in the dried beef samples is shown in Table 1 and Table 2. Recovery values ranged from 114 to 130% for aflatoxin B_1_, B_2_, G_1_, and G_2_. For sterigmatocystin (STE) and OTA, the recovery ranged from 70 to 73%, while HT-2 had 78–81% recovery values. The matrix effects (ME) evaluation was performed using matrix-matched calibration curves; the suppression of the signal (SS) obtained for all the mycotoxins ranged from 16 to 87%. The obtained limits of detection (LOD) and limits of quantification (LOQ) values for each mycotoxin ranged between 0.13 µg/kg (OTA)–127 µg/kg (α-ZEA) and 0.4 µg/kg (OTA)–385 µg/kg (α-ZEA), respectively. For the evaluation of the linearity, calibration curves were constructed for all mycotoxins at twelve different levels of concentrations from 0.9 to 2000 µg/kg for all the mycotoxins. The validation results showed strong correlation coefficients (R^2^ > 0.992). The intra-day accuracy was evaluated by nine determinations at each count level of morning, afternoon, and night on the same day, while inter-day precision was determined for three days. The range of relative standard deviations was between 1.81% and 16.4% for the intra-day precision, and between 2.12% and 15.1% for the inter-day precision. The recovery results ranged from 70% to 130%, except for ochratoxin B (OTB), T-2, 3-ACDON and tenuazonic copper salt (TCS) (36–41, 58–64, 48–57, 11–52) respectively.

### 2.2. Occurrence of Mycotoxins in Dried Beef

The analyzed samples in the current study were contaminated with at least four different types of mycotoxins, while about 60% of samples showed co-occurrence of 4 to 10 different mycotoxins. The occurrence of mycotoxins in the samples are shown in Table 3. The CPA was the most prevalent mycotoxin which was detected in 94% of the samples with an average mean of 51.99 ppb, followed by AFB_1_ with 50%, ranging from 3.91 to 295.41 with an average mean of 95.25 ppb, AFB_2_ has (38% with a range of 0.65–33.13 and an average mean value of 6.93 ppb, AFG_1_ and α-ZEA recording (33%) with a different average mean value of 40.49 and 113.82 ppb, STE (22%) mean value was 5.09 ppb, OTA (16%) has mean value of 0.54 ppb, TCS (14%) has the mean value of 13.58 ppb, OTB (13%) has an average mean value of 0.04 ppb, and AFG_2_ (8%) recorded an average mean value of 2.61 ppb. However, HT-2, β ZEA, CIT, T2, alternariol monomethyl ether (AME), and 3-ACDON were not detected in any of the samples.

### 2.3. Co-Occurence of Mycotoxins in Meats from the Different Sampling Locations

The occurrence based on mean concentration of each mycotoxin are shown according to the different locations in Figure 1. Contamination levels of AFB_1_ were higher and were similar in Ikole and Igede market, followed by α-ZEA, and then AFG_1_ in Igede market. In Ilawe, Aramoko, Otun, Omuo, and Ise/Emure, α-ZEA was the dominant mycotoxin. Ilawe and Ise/Emure samples had the lowest mycotoxin co-contamination of 4 and 6 out of the 18 samples analyzed for mycotoxins, respectively, followed by Ijero and Otun, both with 8 different types of mycotoxins in that order. Ikole and Omuo had 10 different types of mycotoxins that are present in at least one of the samples, this make them the highest locations of simultaneous mycotoxin contaminations, followed by Aramoko, Oye, Ado, and Igede, with each recording 9 different mycotoxins separately. This trend of contamination could be as a result of different fungi peculiarity to the different locations where the cattle were fed and reared, as the meat samples were sourced from about three different states from the Northern part of Nigeria. This could also be influenced by storage, air pollution, and human contact with the product, as well as displaying method. Foods are frequently contaminated with several fungi, and once the temperature and relative humidity is at the peak after contamination, there might be likely mycotoxin production [5]. It has also been established that relatively low water activity (aw < 0.9) with low pH values that is less than 6.0 are principally favorable for fungi growth [26]. The dried meat samples analyzed had low water activity and pH average value of 5.0. This is congruent with favorable water activity and pH values known to favor fungi and mycotoxin production, hence the likely level of fungi contamination and mycotoxin level discovered in the dried meats.

Previous studies showed that mycotoxin contamination in both raw and processed pork meat products is common [15,27]. Although mycotoxins naturally occur in meat primarily as a result of an indirect transfer from naturally contaminated feed [28], production, transportation, storage, and displaying conditions can also influence the occurrence of mycotoxins on dried meat sold in open markets where they are not protected from their environment.

### 2.4. Trichothecenes Level in Dried Beef

Recovery for 3-ACDON and T-2 was low (48–57% and 58–64%) respectively, with the exception of HT-2 which recorded a recovery rate of 78–81%. The other trichothecenes were detected at low concentrations and low occurrences. It was difficult to compare the obtained data with other studies due to the limited literature available.

### 2.5. Ochratoxin A Level in Dried Beef

The result showed that OTA was detected in 14% (n = 108) of the meat samples at concentrations ranging from 0.02 to 3.96 µg/kg, with an average mean of 0.5 µg/kg. OTA contamination levels observed herein are comparable to the one reported by [29], where mean value for OTA in French delicatessen meats was 0.25 µg/kg, but lower than the OTA mean value of 5.23 µg/kg reported by [30] in beef luncheon and 4.55 µg/kg in beef burger samples respectively. This study showed that ochratoxin A levels in dried beef from Ekiti State were lower than the maximum permissible level (1 μg/kg) stipulated for pork products in some European Union (EU) countries [3].

### 2.6. Aflatoxin Total Level in Dried Beef

The total aflatoxin content detected in this study, as show in Table 4 and Figure 2, showed that AFB_1_ values ranged between 3.91 and 295.41 µg/kg with a mean value of 105.4 µg/kg, and AFB_2_ ranged from 0.65 to 33.13 µg/kg with mean value of 6.92 µg/kg. AFG_1_ and AFG_2_ ranged from 2.24 to 257.35 µg/kg and 0.38–19.08 µg/kg with mean value of 40.49 µg/kg and 2.60 µg/kg, respectively. The range and mean value recorded in the present study is higher than the range of 1.10–8.32 µg/kg and 0.15–6.36 µg/kg reported previously [20]. AFB_1_ is the predominant contaminant among the AFs, contaminating about 50% of the beef samples analyzed, which is far higher than the 10% reported in chicken meat from Croatia [9]. It exceeded the maximum permissible level (MPLs) adopted by over 75 countries around the world for AFB_1_ and total aflatoxins of 5 and 10 µg/kg, respectively [3], and not more than 2 and 4 µg/kg stipulated by the European Union for AFB_1_ and total aflatoxins [20].

### 2.7. Other Mycotoxins in the Dried Beef

Other important mycotoxins detected in the meat samples include cyclopiazonic acid, sterigmatocystin, and α-ZEA; interestingly, the concentrations of α-ZEA were relatively higher and ranged from 47.60 µg/kg to 167.34 µg/kg with an average mean value of 113.82 µg/kg in the beef samples across the various selling locations. The actual data and concentration ranges are displayed in Table 4 and Figure 2. Also, high concentrations of cyclopiazonic acid (from 2.11µg/kg to 163.9 μg/kg) with an average mean value of 51.99 µg/kg were similarly detected in the beef samples.

### 2.8. Potential Health Risks for Consumption of Contaminated Dried Beef

Almost all the analyzed beef samples had significant amounts of different mycotoxins, and most people in these areas consume the meat products as a delicacy; as such, consumer’s exposure to different types of mycotoxins is possible. The continuous exposure of consumers to the contaminated dried meat with cyclopiazonic acid (CPA) could result in immunotoxic and hepatotoxic effects that target the muscle, hepatic tissue, and spleen organ of humans [31]. In the case of alpha zearalenone, which is known to be carcinogenic, it causes hormonal imbalance and have effects on reproductive organs [32,33]. The consumption of aflatoxin is known to cause hepatotoxic effects and suppress the immune system, with liver being its target organ [34]. In the case of ochratoxin A, it is carcinogenic, genotoxic, immune-suppressive, nephritic, and causes upper urinary tract diseases, with its target organ being kidney and liver [35]. Sterigmatocystin is genotoxic, cytotoxic, immunotoxic, and carcinogenic, and mainly attacks the liver, immune system, and kidney [36,37]. Tenuazonic acid (copper salt) has been described to exert antiviral, antitumor, antiseptic, cytotoxic, phytotoxic material, and likewise to be highly toxic in living organisms [38].

## 3. Conclusions

In summary, a comprehensive screening for regulated as well as other mycotoxins was conducted in dried beef from Ekiti State for the first time using the modern screening method. Also the efficiency of the LC-MS/MS and extraction solvent employed showed that it was effective for the determination of (CPA, AFB_1,_ AFB_2_, AFG_1_, AFG_2,_ α ZEA, STER, OTA, TCS, and OTB) in dried beef. The analysis revealed that dried beef samples were contaminated with different mycotoxins across the various locations in Ekiti State. Also, the analyzed samples in the current study were contaminated with at least four different mycotoxins, while about 60% of samples showed co-occurrence of 4 to 10 different mycotoxins out of 17 determined mycotoxins. Dried beef samples from Igede and Ikole location had significantly higher mean levels of AFB_1_, followed by AFG_1._ The occurrence or presence of theses mycotoxins and their potential threat to consumer safety should be of great concern. The obtained data offers additional material for discussion about the incidence of mycotoxins in dried beef and their impact on the food safety. Although further research and expansive exposure studies might be necessary, the findings have provided a new outlook for food safety policy makers for the production, handling, storage, and display method for selling dried beef products across the country. It is recommended that further studies be done to know the extent of mycotoxin carry-over into edible tissues of cow when the animals are fed with contaminated feed.

## 4. Materials and Methods

### 4.1. Reagents, Chemicals, and Extraction Kits

Milli-Q quality water (Millipore, M.A.S. 67120 Molsheim, France) was used during the whole analysis. Formic acid, acetonitrile, and methanol were of MS grade, and purchased from Merck (Darmstadt, Germany). The Elisa kits used in the present study were purchased from Sigma-Aldrich (Steinem, Germany), while distilled water was used, and all other chemicals used were of analytical grade.

### 4.2. Standards for LC-MS

The standards of aflatoxin B_1_ (AFB_1_), aflatoxin B_2_ (AFB_2_), aflatoxin G_1_ (AFG_1_), aflatoxin G2 (AFG_2_), ochratoxin A (OTA), ochratoxin B (OTB), sterigmatocystin (STE), a-zearalenol (a-ZEA), ꞵ ZEA zearalenone (ZEN), 3-acetyldeoxynivalenol (3-ADON), alternariol monomethyl ether (AME), Kojic Acid (KA), Citrinin, Sterigmatocystin (STER), Cyclopiazonic acid (CPA), Tenuazonic copper salt (TCS), T-2, and HT-2 toxins were supplied from the National Metrology Institute of South Africa (NMISA). The mycotoxins stock solutions were prepared according to a previous study [25].

### 4.3. Sampling

One hundred and eight dried beef meat were purchased from selected ten major open markets from Ten Local Government Areas of Ekiti State, Nigeria. The markets include Oye, Ilawe, Ise/Emure, Otun, Omuo, Igede, Ikole, Aramoko, and Ado. Sampling locations comprised 2 cities each from 5 geographical zones North, South, East, West, and Central. Each sample was collected from random points of trader’s trays as 12–14 sub-samples (50–100 g) and mixed together to form the bulk sample (200–400 g) and pulverized. A fifty-gram representative sample was obtained from each bulk and stored at −20 °C until analyzed.

### 4.4. Extraction Procedure for LC-MS

The representative samples of particle size between 0.5 and 1 mm were weighed into a 50 mL polypropylene tube (Sarstedt, Nümbrecht, Germany) and enclosed with extraction solvent made up of acetonitrile/water/acetic acid (79:20:1, v/v/v) in 20 mL solvent/ 5g sample. For spiking and recovery experiments, 5 g of samples were used for extraction. Samples were extracted for 90 min at 180 rpm on a Labcon FS16 rotary shaker (Labcon, South Africa), and centrifuged for 15 min at 3500 rpm. The supernatant was carefully transferred into another centrifuge tube and 10 mL of n-hexane was added to breakdown the fat sample [39]. The mixture was vortexed for 2 min and allowed to settle for 20 min at room temperature before the n-hexane phase was removed, and diluted with an equal volume of dilution solvent (acetonitrile/water/ acetic acid 79:20: 1, v/v/v), filtered through a 0. 22 µm syringe nylon filter (Membrane Solutions, Tokyo) into 1.5 mL HPLC vial bottles for injection into the LC-MS/MS system [40].

### 4.5. Instrumentation for LC-MS

A Shimadzu UHPLC instrument LC-MS/MS 8030 equipment (Shimadzu Corporation, Tokyo, Japan), with an ultrafast scan speed of 15,000 u/sec, and a polarity switching of 15 msec was used for the identification and quantification of the analytes. The chromatograph was a LC-30AD Nexera which was connected to a SIL-30 AC Nexera auto sampler and a CTO-20 AC Prominence Column Oven. The oven was equipped with a Raptor^TM^ ARC-18 column from Restek (2.7 µm, 2.1 mm × 100 mm) (Restek Corporation, PA, USA), and maintained at a constant temperature of 40 °C. The mobile phases used consisted of (A) 0.1% formic acid in deionized water and (B) 0.1% formic acid in acetonitrile: methanol (50:50 v/v), and was delivered at a constant flow rate of 0. 2 mL/min with a sample injection volume set at 5 µL. The elution gradient program had a total run time of 20 min, and started with 10% B for 0.01 min and increased steadily to 95% B in 10 min, at which point it was kept constant for 3.5 min, and then the initial condition (10% B) was re-established for 1 min, and the column allowed to re-equilibrate for 5 min for the next run.

Following chromatographic separation, the analytes were committed to a Shimadzu triple quad mass spectrometry detector model 8030 (Shimadzu Corporation, Kyoto, Japan) for detection and quantitation of analyte. The ionization source was an electron spray ionization (ESI) operated in positive mode at an event time of 0.206 sec. Data was acquired by a multiple reaction monitoring (MRM) method at optimized MS conditions for the analyte (Table 1). The interface nebulizing gas flow rate was 3 L/min, DL temperature was 250 °C, heat block temperature was 400 °C, and drying gas flow rate was 15 L/min.

### 4.6. Validation of the Method for LC-MS

The performance characteristics of the method such as matrix effect, linearity, trueness, and intra-day and inter-day precision, limits of detection (LOD) and quantification (LOQ) and selectivity were evaluated.

Matrix effects were estimated using a 12 point matrix-matched calibration curves and a neat standard solution calibration curves within the linear range of 0.9, to 2000 µg/kg. Signal suppression/enhancement (SSE) values, which were calculated by comparing the slopes of matrix-matched calibrations with those of pure standard solution calibrations, were used for the evaluation of the matrix effects [41].

Calibration curves were generated by plotting the area under peak against the concentration of individual mycotoxin in the meat matrices. The method sensitivity was calculated by determining the limit of detection (LOD), which is given as LOD = 3.3 $\times \frac{\mathrm{Conc}}{S/N}$, and limit of quantification (LOQ) given as LOQ = 10 $\times \frac{\mathrm{Conc}.}{S/N}\;$. The LOD and LOQ values of each mycotoxin were determined by mixing the stock standard solutions with the blank matrices.

The recoveries were determined at three different levels using blank samples with nine replicates per concentration level, the spiking was done at low (25 µg/kg), intermediate (50 µg/kg), and high (100 µg/kg) levels of the mycotoxin concentrations. Preparation of samples was carried out in triplicate, as described above. Extraction recovery for each analyte was evaluated by matching the mean peak areas of the samples previously spiked before extraction with that of the samples spiked after extraction.

Precision was evaluated for intra-day and inter-day in non-contaminated samples using standard addition. The samples were spiked with three different concentration levels of each standard (25, 50, and 100 µg/kg) and prepared in triplicate for calculating the intra-day precision, while two samples spiked with 50 and 100 µg/kg of each standard were analyzed daily in triplicate for 5 consecutive days for inter-day precision. The calculation for recovery and precision were based on matrix-matched calibrations. The mycotoxin concentrations (µg/kg) were calculated using calibration curves with mathematical interpolation and multiplication by the sample’s dilution factor.

### 4.7. Statistical Analysis

The concentrations were calculated based on the average recovery values acquired for each analyte. Statistical analysis was done using IBM SPSS version 25 Software with the statistical significance level at (*p* < 0.05).

## Figures and Tables

**Figure 1 toxins-12-00357-f001:**
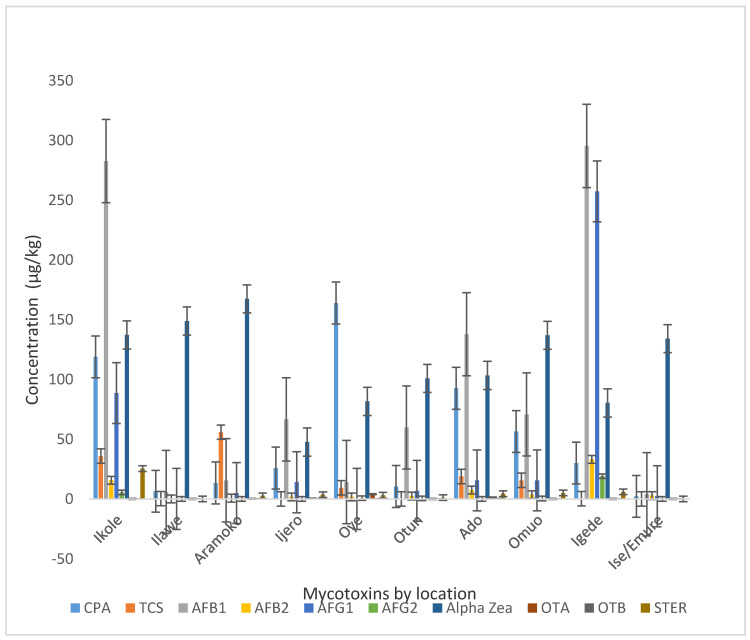
Concentration of mycotoxins in dried beef from different locations/markets.

**Figure 2 toxins-12-00357-f002:**
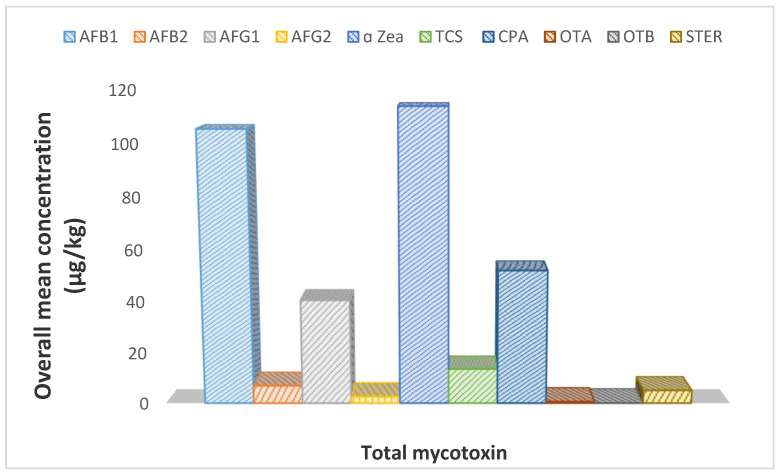
Total mean mycotoxin concentration in analyzed dried beef samples in Ekiti State.

**Table 1 toxins-12-00357-t001:** Multiple reaction monitoring transitions, mass spectrometer conditions, and retention times of the determined mycotoxins.

S/No	Mycotoxin	Ret. Time (min)	Precursor (*mz*)	Products (*mz*)	Q1 Pre Bias (V)	CE	Q3 Pre Bias (V)
1	3ACDON	6.589	338.9	231	−24	−15	−22
				213	−12	−17	−20
2	HT-2	8.817	425	263	−20	−12	−18
				104.9	−16	−47	−19
3	AFG2	7.507	331	245.1	−12	−32	−24
				313	−12	−24	−20
4	AFG1	7.767	329	243	−12	−28	−23
				311.1	−16	−24	−14
5	AFB1	8.25	313	241	−22	−41	−23
				285.1	−22	−24	−29
6	B-ZEA	8.949	323.1	277.2	−16	−16	−18
				305.2	−16	−11	−20
7	a-ZEA	9.415	323.1	277.2	−17	−17	−18
				305.2	−24	−9	−20
8	T-2 Toxin	9.674	467.2	245.1	−13	−11	−16
				305.2	−22	−11	−20
9	ZEA	10.063	319.1	185	−12	−27	−30
				187.1	−15	−21	−19
10	AME	10.125	273	128.1	−10	−49	−21
				115.05	−18	−54	−19
11	OTB	9.331	370.1	205	−13	−22	−21
				324.1	−13	−14	−22
12	OTA	10.132	403.8	239	−15	−27	−24
				221	−12	−38	−21
13	AFB2	8.007	315	259.1	−22	−31	−25
				287	−23	−26	−30
14	TCS	5.02	198.15	125.05	−10	−18	−21
				-	-	-	-
15	Citrinin	5.00	251	205.1	−13	−26	−21
				233	−17	−17	−24
16	STECY	5.01	324.90	310	−22	−24	−30
				281.1	−22	−40	−27
17	CPA	5.01	337.1	196.1	−10	−23	−18
				182.1	−10	−20	−20

3-ADON: 3-acetyldeoxynivalenol, AFG_2_: aflatoxin G_2_, AFG_1_: aflatoxin G_1_, AFB_2_: aflatoxin B_2_, AFB_1_: aflatoxin B_1_, CIT: citrinin, HT-2: HT-2 toxin, T-2: T-2 toxin, α-ZOL: alpha zearalenol, ZEN: zearalenone, OTA: ochratoxin A, STE: sterigmatocystin: AME: Alternariol Methyl Ether.

**Table 2 toxins-12-00357-t002:** Analytical parameters for liquid chromatography coupled with triple quadrupole mass spectrometry (LC-MS/MS) method validation: matrix effect, linearity, limits of detection (LOD), limits of quantitation (LOQ), recoveries at three spiked concentration levels, intra-day, and inter-day precision for the studied mycotoxins.

Mycotoxin	Matrix Effect (%)	Linearity r^2^	LOD µg/kg	LOQ µg/kg	Recovery % Spiked Level (µg/kg)			Intra-day(RSD) % Spiked Level (µg/kg)			Inter-day (RSD) % Spiked Level (µg/kg)		
					25	50	100	25	50	100	1	3	5
AFB_1_	73	0.998	2	6.8	114	121	124	10.9	6.5	4.8	4.22	3.71	3.75
AFB_2_	75	0.999	6	17	121	125	127	4.8	2.7	3.15	3.42	4.35	3.59
AFG_1_	77	0.997	6	16	130	129	130	6.68	6.73	3.07	3.63	4.48	3.69
AFG_2_	83	0.994	44	134	114	120	126	8.2	9.03	3.63	3.57	2.12	3.47
STER	47	0.999	8	23	73	71	71	3.71	6.89	13.1	13.2	15.1	14.7
OTA	60	0.997	0.13	0.4	71	70	70	12.4	5.01	3.29	5.18	5.38	6.81
OTB	48	0.999	0.41	1.2	36	36	41	5.43	5.47	7.33	9.51	5.81	6.46
ZEA	73	0.999	123	374	108	113	114	5.47	5.56	4.7	3.12	5.38	3.1
T-2	52	0.992	84	254	59	64	58	4.64	6.31	3.85	6.19	6.09	6.14
HT-2	35	0.999	3	10	81	79	78	16.4	7.62	4.59	4.86	4.62	4.58
3-ACDON	39	0.997	25	75	48	52	57	2.33	10.1	6.92	6.94	6.18	6.28
AME	66	0.994	55	166	112	114	111	4.41	3.85	2.12	2.34	2.45	2.71
α-ZEA	72	0.998	127	385	117	116	114	3.49	5.54	2.57	3.29	3.55	3.3
β-ZEA	70	0.998	19	57	102	99	99	2.75	4.42	1.81	2.49	3.1	3.37
Citrinin	87	0.998	18	54	101	107	107	2.17	2.6	3.09	4.59	3.33	4.71
CPA	75	0.996	7	20	81	71	71	5.23	4.61	3.15	6.91	6.12	6.11
TCS	43	0.998	8	25	53	26	11	8.39	5.23	5.65	6.38	6.27	6.21

3-ADON: 3-acetyldeoxynivalenol, AFG_2_: aflatoxin G_2_, AFG_1_: aflatoxin G_1_, AFB_2_: aflatoxin B_2_, AFB_1_: aflatoxin B_1_, CIT: citrinin, HT-2: HT-2 toxin, T-2: T-2 toxin, α-ZOL: alpha zearalenol, ZEN: zearalenone, OTA: ochratoxin A, STE: sterigmatocystin: AME: Alternariol Methyl Ether.

**Table 3 toxins-12-00357-t003:** Occurrence (%) of mycotoxins in analyzed samples from different locations (n = 108).

Location	HT 2	CPA	TCS	AFB_1_	AFB_2_	AFG_1_	AFG_2_	ZEA	α ZEA	β ZEA	CIT	T2	OTA	OTB	STER	AME
Ikole	0(15)	14(15)	4(15)	8(15)	9(15)	7(15)	4(15)	0(15)	6(15)	0(15)	0(15)	0(15)	1(15)	3(15)	6(15)	0(15)
Ilawe	0(6)	5(6)	1(6)	1(6)	0(6)	0(6)	0(6)	0(6)	2(6)	0(6)	0(6)	0(6)	0(6)	0(6)	0(6)	0(6)
Aramoko	0(12)	12(12)	1(12)	6(12)	2(12)	7(12)	0(12)	0(12)	5(12)	0(12)	0(12)	0(12)	1(12)	1(12)	1(12)	0(12)
Ijero	0(10)	10(10)	0(10)	7(10)	5(10)	4(10)	0(10)	0(10)	2(10)	0(10)	0(10)	0(10)	5(10)	1(10)	4(10)	0(10)
Oye	0(6)	6(6)	3(6)	4(6)	2(6)	0(6)	1(6)	0(6)	2(6)	0(6)	0(6)	0(6)	3(6)	1(6)	2(6)	0(6)
Otun	0(15)	13(15)	0(15)	4(15)	2(15)	4(15)	1(15)	0(15)	5(15)	0(15)	0(15)	0(15)	0(15)	1(15)	3(15)	0(15)
Ado	0(20)	20(20)	3(20)	12(20)	11(20)	6(20)	0(20)	0(20)	5(20)	0(20)	0(20)	0(20)	6(20)	4(20)	5(20)	0(20)
Omuo	0(14)	13(14)	2(14)	7(14)	5(14)	4(14)	1(14)	0(14)	5(14)	0(14)	0(14)	0(14)	1(14)	2(14)	2(14)	0(14)
Igede	0(6)	6(6)	1(6)	4(6)	3(6)	3(6)	2(6)	0(6)	2(6)	0(6)	0(6)	0(6)	0(6)	1(6)	1(6)	0(6)
Ise/Emure	0(4)	3(4)	0(4)	1(4)	2(4)	1(4)	0(4)	0(4)	2(4)	0(4)	0(4)	0(4)	0(4)	0(4)	0(4)	0(4)
Total %	0%	102 94%	15 14%	54 50%	41 38%	36 33%	9 8%	0%	36 33%	0%	0%	0%	17 16%	14 13%	24 22%	0%

3-ADON: 3-acetyldeoxynivalenol: AFG_2_: aflatoxin G_2_, AFG_1_: aflatoxin G_1_, AFB_2_: aflatoxin B_2_, AFB_1_: aflatoxin B_1_, CIT: citrinin, HT-2: HT-2 toxin, T-2: T-2 toxin, α-ZOL: alpha zearalenol, ZEN: zearalenone, OTA: ochratoxin A, STE: sterigmatocystin: AME: Alternariol Methyl Ether.

**Table 4 toxins-12-00357-t004:** Concentration (µg/kg) of mycotoxins in analyzed dried meat samples from different locations in Ekiti State (n = 108).

Location	Ikole	Ilawe	Aramoko	Ijero	Oye	Otun	Ado	Omuo	Igede	Ise/Emure
HT 2	ND	ND	ND	ND	ND	ND	ND	ND	ND	ND
CPA	118.87 ± 52.05 ^b^	6.42 ± 4.74 ^i^	13.36 ± 5.11 ^g^	25.81 ± 10.54 ^f^	163.93 ± 94.75 ^a^	10.36 ± 8.51 ^h^	92.62 ± 44.61 ^c^	56.43 ± 38.35 ^d^	30.03 ± 14.90 ^e^	2.11 ± 1.11 ^j^
TCS	35.88 ± 28.74	0.30 ± 0.30	55.83 ± 55.83	0	9.27 ± 4.47	0	18.79 ± 11.19	15.6 ± 14.93	0.16 ± 0.16	0
AFB_1_	282.75 ± 136.66	5.85 ± 5.85	15.6 ± 9.62	66.57 ± 45.71	14.1 ± 8.87	59.81 ± 58.53	137.87 ± 60.05	70.69 ± 51.14	295.41 ± 272.73	3.91 ± 3.91
AFB_2_	15.69 ± 7.32 ^ab^	0	0.65 ± 0.47 ^b^	1.72 ± 0.67 ^b^	1.69 ± 1.24 ^b^	2.5 ± 2.31 ^b^	7.42 ± 2.88 ^b^	3.6 ± 2.51 ^b^	33.13 ± 31.02 ^a^	2.85 ± 2.48 ^b^
AFG_1_	88.66 ± 39.09 ^b^	0	4.94 ± 1.91 ^c^	13.95 ± 9.96 ^b^	0	6.82 ± 5.78 ^c^	15.45 ± 8.21 ^b^	15.56 ± 8.81 ^b^	257.35 ± 208.95 ^a^	2.24 ± 2.24 ^d^
AFG_2_	5.53 ± 2.55 ^b^	0	0	0	0.64 ± 0.64 ^b^	0.45 ± 0.45 ^b^	0	0.38 ± 0.38 ^b^	19.08 ± 14.87 ^a^	0
ZEA	ND	ND	ND	ND	ND	ND	ND	ND	ND	ND
α Zea	137.2 ± 49.00	148.89 ± 124.93	167.34 ± 70.53	47.6 ± 39.14	81.64 ± 51.73	100.83 ± 44.62	103.35 ± 44.46	136.96 ± 52.71	80.33 ± 50.97	134.11 ± 77.59
β Zea	ND	ND	ND	ND	ND	ND	ND	ND	ND	ND
CIT	ND	ND	ND	ND	ND	ND	ND	ND	ND	ND
T2	ND	ND	ND	ND	ND	ND	ND	ND	ND	ND
OTA	0.02 ± 0.02 ^b^	0	0.03 ± 0.03 ^d^	0.28 ± 0.10 ^c^	3.96 ± 3.60 ^a^	0	1.12 ± 0.64 ^b^	0.02 ± 0.02 ^d^	0	0
OTB	0.03 ± 0.02	0	0.01 ± 0.05	0.03 ± 0.03	0.03 ± 0.03	0.01 ± 0.00	0.22 ± 0.19	0.01 ± 0.00	0.01 ± 0.01	0
STER	25.37 ± 13.77	0	2.64 ± 2.64	3.56 ± 1.78	3.06 ± 2.38	1.08 ± 0.59	4.36 ± 3.42	4.98 ± 4.62	5.89 ± 5.89	0
AME	ND	ND	ND	ND	ND	ND	ND	ND	ND	ND
3-AC DON	ND	ND	ND	ND	ND	ND	ND	ND	ND	ND

Letters with different superscript along the same row indicate significant difference, Concentration value is the mean of all the samples from the same location, Letters with the same superscript along the same row indicate no significant difference, ND indicates not detected, *p* = (<0.05).

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
