# Peer review of "Multi Mycotoxin Determination in Dried Beef Using Liquid Chromatography Coupled with Triple Quadrupole Mass Spectrometry (LC-MS/MS)"

_toxins, 2020, doi:10.3390/toxins12060357_

Round 1

Reviewer 1 Report

I find the manuscript interesting... but the authors must improve it for the publication on Toxins.

These are my suggestions:

Lines 28-79:

The authors must update the references in the introduction. They cite papers up to 20 years ago.

Lines 81-111:

check typos in the title (line 81) and (Table 1 and 2, Line 83).

The authors must specify also in this paragraph the spiking levels in the recovery tests as well as the method of calculation of LOD and LOQ (It is reported at line 295-296 but it seems confused 3.3? con or conc.? I do not understand)

Moreover: why do the authors include FBs, DON and 15ADON if they are not able to extract them with their method?  They must re-organize the manuscript because they developed a method for the detection of 17 and not 22 mycotoxins (and some of them, such as OTB, have not good recoveries).

Lines 113-125

Check typo line 115 (60%)

The authors did not detect 15-ADON… but they are not able to recover it (as well as FBs and DON, not reported correctly in the table)

Lines 126-154

Check typo line 131 (18)

Too old references (Line 141-142-152): there are many recent papers in literature with deepen studies about the considerations of the authors reported in this paragraph (for many matrices and many mycotoxins)

Lines 155- 161

Check typo line 155 .

The authors did not investigate about DON and 15ADON (as they wrote) because they have not developed a method able to extract them!!!

Lines 169-170

This part lacks of references. The authors must add them (in particular when they cite EU and Italian law about OTA)

Lines 180-181

I do not understand what the authors want to speculate. Please re-phrase.

Lines 184-187

Table 3 (and Figure 1 and 2 as well) lack of statistics. In particular the authors write about significant differences but they do not specify what kind of test they did and what is the P. What is reported at line 310-315 is not sufficient

Lines 195-210

The considerations are good but not complete. The references are too old and some informations needs an update (ie Tenuazonic acid is recently discovered as a potential Anti-Alzheimer mutlitarget scaffold!)

Line 298-303

Considering LOQ/LOD and the levels found of the majority of the mycotoxins (such as AFs, OTA and so on) the lower level of recovery tests seems unadequate.

Author Response

Response to Reviewer 1 Comments

Point 1: Lines 28-79: The authors must update the references in the introduction. They cite papers up to 20 years ago.

Response 1: due to the scanty literature on beef mycotoxin, the introduction cannot but reflect some old references that speak to the subject matter. However, few were changes without much alteration to the sentences. For instance, Line 35 has been updated to a more recent reference and it now reflects as (Duarte et al., 2010a; Magan et al., 2011).

Point 2: Lines 81-111: check typos in the title (line 81) and (Table 1 and 2, Line 83). The results obtained from the validation of the extraction and LC-MS method used for the determination of the mycotoxins in the dried beef samples are shown in (Table 1 and 2).

Response 2: The results obtained from the validation method used in sample extraction and LC-MS method in the determination of mycotoxins in the dried beef samples is shown in (Table 1 and 2). Line 82-83

Point 3: The authors must specify also in this paragraph the spiking levels in the recovery tests as well as the method of calculation of LOD and LOQ (It is reported at line 295-296 but it seems confused 3.3? con or conc.? I do not understand)

Response 3: The recoveries were clearly stated in Line 297-299 that it was determined at three different levels using blank samples with nine replicates per concentration level, the spiking was done at low (25 µg/kg), intermediate (50 µg/kg) and high (100 µg/kg) levels of the mycotoxin concentrations. In-Line 294, the formula has been corrected to 3.3 conc.

Point 4: Moreover: why do the authors include FBs, DON and 15ADON if they are not able to extract them with their method?  They must re-organize the manuscript because they developed a method for the detection of 17 and not 22 mycotoxins (and some of them, such as OTB, have not good recoveries).

Response 4: Initially, we were hoping that the extraction method will be able to extract FBs, DON and 15 ADON, that’s why we reported the mycotoxins. However, it has been removed from the script since the recovery was bad.

Point 5: Check typo line 115 (60%)

Response 5: The typo has been corrected to reflect (The analyzed samples in the current study were contaminated with at least four different types of mycotoxins. While about 60% samples showed co-occurrence of 4 to 10 different mycotoxins). Line 114-115

Point 6: The authors did not detect 15-ADON… but they are not able to recover it (as well as FBs and DON, not reported correctly in the table) Lines 126-154

Response 6: 15-ADON, FBs and DON, has been deleted from table 4) Lines 126-154

Point 7: Check typo line 131 (18)

Response 7: The typographical error observed on line 131 has been corrected to reflect (Ilawe and Ise/Emure samples had the lowest mycotoxin co-contamination of 4 and 6 out of the 18 samples analyzed for mycotoxins respectively. Line 132

Point 8: Too old references (Line 141-142-152): there are many recent papers in literature with deepening studies about the considerations of the authors reported in this paragraph (for many matrices and many mycotoxins)

Response 8: The old references mentioned such as (Styriak et al., 1998) has been changed to (Magan et al., 2011) line 141-142 and Gareis and Scheuer (2000) line 152 was changed to (Völkel et al., 2011) and under reference in line 415

Point 9: Lines 155- 161 Check typo line 155. The authors did not investigate about DON and 15ADON (as they wrote) because they have not developed a method able to extract them!!!

Response 9: DON and 15ADON (as written) but not developed and able to extract has been deleted from line 156-159 now as pointed out. Likewise, the typographical error has been corrected to reflect (It was difficult to compare the obtained data with other studies due to the limited number of literature available) Line 158-159

Point 10: Lines 169-170. This part lacks references. The authors must add them (in particular when they cite EU and Italian law about OTA)

Response 10: This study showed that ochratoxin A level in dried beef  from Ekiti State were lower than the maximum permissible level (1 μg/kg) stipulated for pork products in some EU countries (Pleadin et al., 2015) Line 166-168

Point 11: Lines 180-181 I do not understand what the authors want to speculate. Please re-phrase.

Response 11: The sentence has been re-phrased to read ‘’It exceeded the maximum permissible level (MPLs) adopted by over 75 countries around the world for AFB1 and total aflatoxins of 5 and 10 µg/kg respectively (Pleadin et al., 2015) and not more than 2 and 4 µg/kg stipulated by European Union for AFB1 and total aflatoxins (Herzallah, 2009).’’ Line 177-180

Point 12: Lines 184-187 Table 3 (and Figure 1 and 2 as well) lack of statistics. In particular, the authors write about significant differences but they do not specify what kind of test they did and what is the P. What is reported at line 310-315 is not sufficient

Response 12: Table 3 has been updated in Line 184. While the statistical analysis was done using IBM SPSS version 25 Software, this is well stated in Line 315. The P-value has been included in line 316.

Point 13: Lines 195-210 the considerations are good but not complete. The references are too old and some information’s needs an update (i.e. Tenuazonic acid is recently discovered as a potential Anti-Alzheimer multi-target scaffold!)

Response 13: Malekinejad et al., 2011 has been added to the reference in line 200 Tenuazonic acid Tenuazonic acid

Point 14: Line 298-303 Considering LOQ/LOD and the levels found of the majority of the mycotoxins (such as AFs, OTA and so on) the lower level of recovery tests seems inadequate.

Response 13: The LOQ/LOD of the majority of the mycotoxins were reported as found

Reviewer 2 Report

The article entitled " multi mycotoxin determination in dried beef using liquid chromatography coupled with triple quadrupole mass spectrometry (LC-MS/MS)" describes a study to evaluate the simultaneous determination of 22 mycotoxins in dried beef.

The paper is well organized and presents essential and necessary data from a scientific point of view. The performance characteristics of a LC-MS/MS procedure offers an efficient analytical approach for multi-residue analysis of mycotoxins, even at low regulated levels for foods. In addition, the occurrence of these mycotoxins in dried beef, from Ekiti State markets, is an indication of possible exposure of its consumers to the dangers of mycotoxin that is usually associated with severe health problems.

In my opinion, the article can be published after Minor Revision.

Comments

  • I would like the meat production and transformation process to be described in detail, including with a flow diagram. In addition, that the authors indicate if there is a Quality Assurance System applied, such as HACCP, in the process.

  • In addition, in section 2.8. of results, “Potential health risks….” The authors could make a deterministic estimation of the risk for the population with the data obtained in the present study and the consumption data for dried beef in the population.

Author Response

Response to Reviewer 2 Comments

Point 1: I would like the meat production and transformation process to be described in detail, including with a flow diagram. In addition, the authors indicate if there is a Quality Assurance System applied, such as HACCP, in the process.

Response 1: Inline 71-73, the meat production and transformation process have been briefly described. This is to avoid major changes to the manuscript. Unfortunately, there is no assurance system applied since the process is traditional, neither is HACCP in the process. This is one of the reasons why this research was undertaken to enlighten the people who are into its processing as well as inform the necessary authorities of the inherent dangers associated with the traditional way of processing the beef.

Point 2: In addition, in section 2.8. of results, “Potential health risks….” The authors could make a deterministic estimation of the risk for the population with the data obtained in the present study and the consumption data for dried beef in the population

Response 2: Initially, the authors planned to make a deterministic estimate of the mycotoxin risk, but had a challenge with consumption date which is not available. Hence the authors resulted in the speculative risk estimation as seen in the current research.

However, the authors are now working on the collection of consumption data of dried beef in the regions where samples were collected, to determine and estimate the mycotoxin risk for the population with data obtained.

Reviewer 3 Report

The quality of the submitted manuscript is quite high and appropriate for submission to Toxins. The main question is the novelty of the study in comparison to the previous research and the standard procedure for determination of mycotoxins in meat. 

Author Response

Response to Reviewer 3 Comments

Point: The quality of the submitted manuscript is quite high and appropriate for submission to Toxins. The main question is the novelty of the study in comparison to the previous research and the standard procedure for the determination of mycotoxins in meat. 

Response: according to the reviewer, the submitted manuscript quality is high and appropriate for submission to Toxins.  The most important findings of this study is that mycotoxin is been reported for the first time on dried beef sold in Ekiti State at different locations, and also using a validated method for the LC-MS that can simultaneously determine several mycotoxins in dried beef.  

Reviewer 4 Report

This research work describes the analyses of several mycotoxins in dried beef meat from different origins, pointing out the high diversity of mycotoxins that it is possible to find in this type of products. The works requires some corrections, improvements and clarifications before getting published: 

  • Line 49: "although" does not seem to be the proper adverb.
  • Line 53: "For instance" should be the beginning of a new sentence, otherwise is difficult to follow. 
  • Line 56: "susitable" has to be changed to "suitable".
  • Line 83: no parenthesis in Tables.
  • Line 141: "established" lacks the "a".
  • Line 160: "comparison" is not properly written.
  • Lines 169-170: references are needed for the legal values.
  • Line 187: "references" misses some letters.
  • Line 198: "consumers" is not properly written. 
  • Results: a more logical order would be: 1) total content; 2) specific mycotoxins; 3) co-ocurrence. 
  • Discussion should include strategies to avoid meat contamination. 
  • Tables: please, include EU maximum levels in tables to compare and discuss more about if the levels found are above or below those values. 
  • M&M: sampling must be the first subsection, subsections 4.1. and 4.2. can be merged. 
  • Figure 1: please explain about the huge SD. Are these values normal in this type of analyses?
  • Statistical analyses must be explained in detail in M&M (ANOVA...) and what is significant needs to be commented in the text.
  • Subsection 4.7.: how the mycotoxins concentrations were calculated is not part of the statistical analyses, please move this together to the description of mycotoxins determination. 
  • The sections "author contributions", "funding" and "conflit of interests" must be fullfilled. 

Author Response

Response to Reviewer 4 Comments

Point 1: Line 49: "although" does not seem to be the proper adverb.

Response 1: although has been changed to while inline 49

Point 2: Line 53: "For instance" should be the beginning of a new sentence, otherwise is difficult to follow. 

Response 2: For instance" has been changed to reflect the beginning of a new sentence in Line 53. 

Point 3: Line 56: "susitable" has to be changed to "suitable".

Response 3: "susitable" has been changed to "suitable".      

Point 4: Line 83: no parenthesis in Tables.

Response 4: The tables now have parenthesis

Point 5: Line 141: "established" lacks the "a".     

Response 5: line 140, the word has been corrected to include the omitted ‘’a’’

Point 6: Line 160: "comparison" is not properly written.

Response 6: The entire sentence has been reconstructed line 160-167

Point 7: Lines 169-170: references are needed for the legal values.

Response 7: This study showed that ochratoxin A level in dried beef  from Ekiti State were lower than the maximum permissible level (1 μg/kg) stipulated for pork products in some EU countries (Pleadin et al., 2015) Line 166-168

Point 8: Line 187: "references" misses some letters.

Response 8: the whole sentence has been removed as it serves no purpose

Point 9: Line 198: "consumers" is not properly written. 

Response 9: Consumers has been properly written

Point 10: Results: a more logical order would be: 1) total content; 2) specific mycotoxins; 3) co-occurrence. 

Response 10: I believe the order is okay, the reviewer only thinks that it could be more logical in the suggested manner

Point 11: Discussion should include strategies to avoid meat contamination. 

Response 11: the authors are also looking at total assessment of dried beef production to identify the critical controls of the production process.

Point 12: Tables: please, include EU maximum levels in tables to compare and discuss more about if the levels found are above or below those values. 

Response 12: This were adequately taken care of at 2.5 and 2.6. it was properly referred as well during the result discussions.

Point 13: M&M: sampling must be the first subsection, subsections 4.1. And 4.2. Can be merged. 

Response 13: I think it’s okay the way it is for clarity. 4.1 talks about Reagents, chemicals and Extraction Kits while 4.2 clearly spelt out the standard used for LC-MS

Point 14: Figure 1: please explain the huge SD. Are these values normal in this type of analyses?

Response 14: This is the first time the extraction method is been used for multi-mycotoxin from dried beef. This SD difference could be as a result of meat matrix which is more of protein

Point 16: Statistical analyses must be explained in detail in M&M (ANOVA...) and what is significant needs to be commented in the text.

Response 16: This has been properly captured inline 307-307

Point 17: Subsection 4.7.: how the mycotoxins concentrations were calculated is not part of the statistical analyses, please move this together to the description of mycotoxins determination. 

Response 17: it has been moved to line 305-306 as suggested

Point 18: The sections "author contributions", "funding" and "conflict of interests" must be fulfilled. 

Response 18: The authors’ contribution, funding and conflict of interest has been included from line 311-316